# The Current and Prospective Adjuvant Therapies for Hepatocellular Carcinoma

**DOI:** 10.3390/cancers16071422

**Published:** 2024-04-06

**Authors:** Jeng-Shiun Du, Shih-Hsien Hsu, Shen-Nien Wang

**Affiliations:** 1Division of Hematology and Oncology, Department of Internal Medicine, Kaohsiung Medical University Hospital, Kaohsiung 80756, Taiwan; ashiun@gmail.com; 2Center for Cancer Research, Kaohsiung Medical University, Kaohsiung 80708, Taiwan; 3Graduate Institute of Medicine, College of Medicine, Kaohsiung Medical University, Kaohsiung 80708, Taiwan; 4Research Center for Environmental Medicine, Kaohsiung Medical University, Kaohsiung 80708, Taiwan; 5Division of General and Digestive Surgery, Department of Surgery, Kaohsiung Medical University Hospital, Kaohsiung 80708, Taiwan; 6Department of Surgery, Faculty of Medicine, Kaohsiung Medical University, Kaohsiung 80708, Taiwan

**Keywords:** hepatocellular carcinoma (HCC), adjuvant therapy, recurrence, improvement

## Abstract

**Simple Summary:**

Recently, significant breakthroughs have been made in the medical treatment of advanced hepatocellular carcinoma (HCC). There is no consensus on whether adjuvant therapy is necessary after curative surgery for HCC. However, some patients still face a high risk of recurrence. Consequently, researchers have been exploring adjuvant therapies, such as chemotherapy, targeted therapy, and immunotherapy, to mitigate the risk of recurrence and improve patient prognosis. Although there is ongoing debate regarding the necessity of adjuvant therapy after surgery, examining the literature and investigating risk factors for high recurrence and the efficacy of current treatments can provide clinicians with more comprehensive information to develop individualized treatment plans. Moreover, advancements in future research and clinical practice will enhance our understanding and treatment of high recurrence in HCC.

**Abstract:**

Hepatocellular carcinoma (HCC) stands as the most prevalent form of primary liver cancer and is highly invasive and easily recurs. For HCC, chemotherapy shows limited effect. The gold standard for HCC treatment includes curative surgical resection or liver transplantation. However, the recurrence rate at 5 years after liver resection is estimated at approximately 70% and even at 5 years after liver transplantation, it is 20%. Therefore, improving survival outcomes after curative surgical resection of liver cancer is crucial. This review highlights the importance of identifying risk factors for HCC recurrence following radical surgical resection and adjuvant therapy options that may reduce the recurrence risk and improve overall survival, including local adjuvant therapy (e.g., transcatheter arterial chemoembolization and radiotherapy), adjuvant systemic therapy (e.g., small molecule targeted therapy and immunotherapy), and other adjuvant therapies (e.g., chemotherapy). However, further research is needed to refine the use of these therapies and optimize their effectiveness in preventing HCC recurrence.

## 1. Introduction

Hepatocellular carcinoma (HCC) is the seventh most common malignancy worldwide and ranks fifth in Taiwan. HCC contributed to most liver cancer deaths, and it was also the second-highest cause of cancer-related deaths in Taiwan in 2016 [1,2]. In 2018, an estimated incidence rate of 9.3 cases of liver cancer per 100,000 person-years was diagnosed, with a corresponding mortality rate of 8.5 per 100,000 person-years [1,2]. The prognosis for HCC is poor, and curative surgery is considered one of the optimal strategies for HCC treatment. Recent progress in immunotherapy has improved outcomes for patients with advanced HCC. However, the recurrence rate after curative resection remains high, and there are limited studies on adjuvant therapy for HCC. This article reviews the pathogenesis and risk factors of HCC, and the role of targeted therapy, radiotherapy (RT), transcatheter arterial chemoembolization (TACE), and immunotherapy in patients with HCC after radical surgical resection.

## 2. Risk Factors

### 2.1. Nature of Liver Cancer Etiology

Liver cancer, particularly HCC, is a significant global health concern. Its incidence rate has increased in many countries over the past few decades. Hepatitis B virus (HBV) and hepatitis C virus (HCV) are currently the most important global risk factors for HCC. Due to factors such as HBV vaccination and improved treatments for chronic infections, the prevalence of these viruses is expected to decline. HBV vaccination, especially in newborns, has shown positive effects in some countries, with observable benefits in young adults. Globally, the prevalence of metabolic risk factors for HCC is increasing. Conditions such as metabolic syndrome, obesity, type II diabetes, and non-alcoholic fatty liver disease (NAFLD) are emerging as major contributors to liver cancer. Additionally, excessive alcohol consumption remains a significant and challenging risk factor for liver cancer. Aflatoxin contamination of food crops in specific regions also poses a risk.

### 2.2. Risk Factors for HCC Recurrence

Yao et al. [3] utilized a large-scale multicenter database from China to investigate risk factors, recurrence patterns, and treatment outcomes among patients who underwent curative hepatic resection (HR) for early-stage HCC at BCLC stage 0/A. Among the 1424 patients in the study, 47.7% (679 patients) experienced recurrence during a median follow-up period of 54.8 months. Recurrences were categorized into early recurrence (within 2 years after surgery) and late recurrence (2 years or more after surgery). Early recurrence occurred in 60.1% of cases, while late recurrence was observed in 39.9% of cases. Several independent risk factors for postoperative recurrence were identified, including cirrhosis, an alpha-fetoprotein level exceeding 400 ug/L before the operation, a tumor size greater than 5 cm, the presence of multiple tumors, satellites, microvascular invasion (MVI), and the need for intraoperative blood transfusion [3,4,5].

A scoring system based on tumor characteristics was developed to predict the efficacy of TACE adjuvant therapy. The system included factors such as MVI, incomplete tumor envelope, tumor diameter, number of tumors, and surgical margin [6]. Identifying risk factors for recurrence can help identify high-risk patients who may benefit from future adjuvant therapies.

## 3. Adjuvant Local Treatments

### 3.1. Transcatheter Arterial Chemoembolization

TACE is highlighted as one of the main treatment methods for locally progressive, unresectable HCCs that are not suitable for liver transplantation (LT). It is also gaining interest as an adjuvant treatment to prevent the recurrence of HCC after surgical resection. However, the benefits of adjuvant TACE therapy after hepatectomy are inconsistent across existing studies.

Several randomized controlled trials (RCTs) from Japan and China supported using postoperative adjuvant TACE therapy to reduce recurrence rates and improve overall survival (OS) in patients at moderate to high risk of recurrence [7]. Most of these studies have demonstrated significantly lower recurrence rates, longer recurrence-free survival (RFS) or disease-free survival (DFS), and improved OS in patients receiving adjuvant TACE after radical hepatectomy [8,9,10,11].

A meta-analysis [12] involving 11,165 patients showed a statistically significant improvement in OS and DFS for those who received adjuvant TACE compared to those who underwent curative resection alone. Subgroup analysis indicated that postoperative TACE may particularly benefit patients at high risk of recurrence, such as those with tumor diameters greater than 5 cm, positive MVI, or multinodular tumors.

In patients at low risk of recurrence, a retrospective study suggested that prophylactic TACE interventions are less effective in reducing recurrence [13,14]. A meta-analysis and systematic review indicated that patients with low-risk recurrent HCC may not benefit from adjuvant TACE therapy [14].

A meta-analysis of 11 studies [15] demonstrated that adjuvant TACE offers superior 1-, 3-, and 5-year OS and DFS rates compared to HR alone. Furthermore, adjuvant TACE could particularly benefit HCC patients with MVI who have a tumor diameter greater than 5 cm or multinodular tumors. Liver resection (LR) combined with portal vein tumor thrombus (PVTT) removal and postoperative TACE is suggested to be beneficial. In a retrospective cohort, postoperative adjuvant TACE was associated with longer OS, especially in patients with a more significant extent of type II or III PVTT [16,17,18,19,20]. Emerging data suggest the prospect of various modalities in managing HCC with PVTT, including RT, TACE, transarterial radioembolization (TARE), and hepatic artery infusion chemotherapy (HAIC). Combinations of drugs and modalities appear effective for both systemic and loco-regional control. Advances in therapeutic options and refined delivery systems allow for the down-staging of patients, making them eligible for curative resection, including LR and even LT. Adjuvant therapy and prompt management of recurrences are emphasized as crucial for prolonging OS and DFS. However, adjuvant TACE is needed for prospective trials to address the evolving landscape and determine optimal management strategies for HCC patients with PVTT [21].

Adjuvant TACE is considered safe and effective for Asian patients with HCC at a high risk of recurrence. It may be an effective treatment for preventing tumor recurrence and metastasis after surgical resection of early to mid-stage HCC. Variations in reports concerning the population, treatment protocol, timing, and course of adjuvant TACE warrant further clinical exploration. The studies of adjuvant transarterial chemotherapy are provided in Table 1.

### 3.2. Hepatic Arterial Infusion Chemotherapy and Portal Vein Infusion Chemotherapy (PVC)

#### 3.2.1. HAIC in Postoperative Adjuvant Treatment

For patients with multiple tumors combined with MVI, another study reported a higher OS in the HAIC group compared to the surgery-alone group [11]. A retrospective study involving 85 patients in China evaluated the impact of postoperative HAIC on the survival probability of patients who underwent radical hepatectomy for HCC. In another study involving 42 patients who underwent two sessions of HAIC following radical hepatectomy, the results demonstrated several positive outcomes compared to those of the control group. The HAIC group exhibited a significantly higher 5-year intrahepatic RFS probability, a lower risk of intrahepatic recurrence, a considerably higher 5-year DFS probability, and a significantly higher OS probability.

Specifically, the hazard ratio (HR) for intrahepatic recurrence in the HAIC group was 0.5615 (95% CI: 0.3234–0.9749) compared to the control group, indicating a statistically significant reduction in the risk of intrahepatic recurrence. Similarly, the HR for DFS in the HAIC group was 0.591 (95% CI: 0.3613–0.9666), and for OS, it was 0.5768 (95% CI: 0.3469–0.9589). These HR values indicate a significantly higher probability of long-term survival in the HAIC group.

These findings suggest that postoperative HAIC using a combination of 5-fluorouracil, oxaliplatin, and mitomycin-C is effective and safe in reducing intrahepatic recurrence and improving the OS in patients with HCC after radical hepatectomy. The study demonstrates the potential benefits of this adjuvant therapy approach in managing HCC following surgical resection. However, it is important to note the small sample size of the study; further research with larger cohorts is needed to validate these results and evaluate the long-term efficacy and safety of HAIC.

#### 3.2.2. Portal Vein Infusion Chemotherapy in Postoperative Adjuvant Treatment for HCC with Portal Vein Tumor Thrombosis

In a retrospective study [22] involving cases of hepatocellular carcinoma (HCC) with portal vein tumor thrombosis (PVTT), it was observed that the postoperative adjuvant portal vein chemotherapy (PVC) group exhibited a notably extended median time to recurrence (TTR) and overall survival (OS) compared to the control group. Moreover, the cumulative recurrence rate was significantly reduced in the PVC group. Additionally, Hamada et al. reported improved disease-free survival (DFS) and OS in patients with HCC and concurrent portal infiltration who received adjuvant hepatic arterial infusion chemotherapy (HAIC) compared to those who did not undergo HAIC treatment.

#### 3.2.3. Meta-Analysis and Randomized Controlled Trial

An analysis of 11 retrospective cohort studies indicated that the use of adjuvant hepatic arterial infusion chemotherapy (HAIC) following surgical resection resulted in enhanced overall survival (OS) and disease-free survival (DFS) when compared to surgical treatment alone [11]. The inclusion of multiple studies in a meta-analysis allows for a larger sample size, providing a more robust analysis of the effectiveness of adjuvant HAIC. The retrospective cohort design of these studies involved analyzing data from patients who had already undergone surgical resection for HCC, comparing outcomes between those who received adjuvant HAIC and those who did not.

The pooled analysis of these studies indicated that adjuvant HAIC was associated with improved OS and DFS. This means that patients who received HAIC after surgical resection had a higher probability of long-term survival and a lower risk of disease recurrence compared to those who underwent surgery alone.

It is worth noting that retrospective studies have inherent limitations, including potential biases and confounding factors that may affect the results. Therefore, while these findings are promising, further prospective studies or RCTs are needed to provide more robust evidence on the effectiveness and safety of adjuvant HAIC after surgical resection for HCC.

In a prospective, open-label, phase III RCT involving 127 patients, the use of postoperative transarterial infusion chemotherapy with the FOLFOX regimen as adjuvant therapy for HCC with MVI was found to significantly prolong both OS and DFS when compared to the postoperative observation group [11]. The study design involved randomly assigning patients into two groups. The first group received postoperative transarterial infusion chemotherapy using the FOLFOX regimen, which consists of fluorouracil, leucovorin, and oxaliplatin. The second group, the control group, underwent postoperative observation without additional treatment. The results demonstrated that patients in the transarterial infusion chemotherapy group had significantly prolonged OS and DFS compared to those in the observation group. This implies that using adjuvant transarterial infusion chemotherapy with the FOLFOX regimen effectively improved long-term survival and reduced the risk of disease recurrence in patients with HCC and MVI.

It is important to consider the limitations of this study, such as potential bias and the relatively small sample size. Further research with larger cohorts, ideally in the form of multi-center RCTs, would be necessary to confirm these findings and provide more robust evidence of the benefits of postoperative transarterial infusion chemotherapy in this patient population.

Both HAIC and PVC have shown potential benefits as adjuvant treatments for HCC following surgical resection, with reported improvements in RFS, OS, and DFS. However, additional research, particularly in larger prospective trials with longer follow-up periods, is necessary to strengthen the evidence and validate these findings.

### 3.3. Radiotherapy

Adjuvant RT may play a role in specific subgroups of HCC patients, such as those with narrow margins, small tumor sizes, combined MVI, or PVTT. The outcomes vary among studies, which highlights the importance of individualized treatment approaches and the need for further research to confirm these findings [23].

In a prospective RCT [24] with 10-year real-world evidence, the study examined the viability and effectiveness of adjuvant radiotherapy in patients with central hepatocellular carcinoma (HCC) after narrow-margin hepatectomy (<1 cm). The results showed no notable discrepancy in recurrence-free survival (RFS) and overall survival (OS) between the groups treated with adjuvant radiotherapy and the control groups. Nevertheless, patients with small HCC (5 cm) demonstrated a significantly longer RFS, and those with small HCC exhibited a significantly improved OS compared to the control group during the period of 2 to 5 years post-treatment.

A single-arm prospective phase II trial [25] investigated postoperative intensity-modulated radiotherapy (IMRT) as an adjuvant treatment for HCC following narrow-margin (<1 cm) hepatectomy. This study enrolled 76 eligible patients who underwent narrow-margin resection, received adjuvant radiotherapy, and were evaluated for OS, DFS, recurrence patterns, and toxicity. The trial reported the following favorable outcomes: a 3-year OS rate of 88.2%, a 3-year DFS rate of 68.1%, and a 5-year OS rate of 72.2%. Importantly, the trial noted that no marginal recurrence was observed in the study cohort, suggesting that adjuvant RT may have effectively prevented the recurrence of the disease at the surgical margins.

## 4. Adjuvant Systemic Therapy

There has been some progress in systemic treatment with novel targeted drugs in phase III studies for advanced HCC. In a meta-analysis published in the *American Journal of Translational Research* in April 2021, Antonio Facciorusso and colleagues compared the efficacy of lenvatinib and sorafenib as first-line treatments for advanced hepatocellular carcinoma (HCC) patients [26]. The analysis, which included five studies involving 1481 patients, found no significant difference in overall survival between the two treatments (hazard ratio [HR] 0.81, 95% confidence interval [CI] 0.58–1.11). The median survival was 13.4 months (CI 9.38–17.48) for lenvatinib and 11.4 months (CI 8.46–14.47) for sorafenib recipients. Lenvatinib showed a statistically significant improvement in progression-free survival (PFS) compared to sorafenib (HR 0.67, CI 0.48–0.94) and higher rates of objective response (33.3%, CI 23.6–43% vs. 6.5%, CI 3.5–9.5%; odds ratio [OR] 7.70, CI 2.99–19.82) and disease control rate (76.9%, CI 70.4–83.5% vs. 52.7%, CI 40.7–64.6%; OR 2.41, CI 1.55–3.77) compared to sorafenib.

These advancements in targeted therapeutics represent promising developments in the treatment of HCC, offering options for both first-line and second-line interventions and potential improvements in adjuvant therapy to help prevent cancer recurrence. The use of targeted drugs in the adjuvant setting indicates ongoing efforts to explore new treatment approaches beyond initial cancer management phases.

### 4.1. Target Therapy

#### 4.1.1. Sorafenib

Sorafenib demonstrates antitumor activity in advanced HCC and superior OS, but there is less evidence in postoperative adjuvant therapy. The STORM study [27], a randomized, double-blind, placebo-controlled phase III clinical trial, evaluated the efficacy of sorafenib as an adjuvant therapy for patients with resected HCC. It enrolled 1114 patients randomly divided into sorafenib treatment or placebo groups. After a median treatment duration of about 12.5 months, no statistically significant difference was observed concerning RFS between the two groups. Additionally, the sorafenib group experienced a poor adverse event profile, including four treatment-related deaths. Some believe the negative result of the STORM study could be attributed to ineffective patient selection at a high risk of recurrence. Furthermore, in the BIOSTORM study [28], hepatocytic protein kinase RNA-like endoplasmic reticulum kinase (PERK) and MVI were independent poor prognostic factors of RFS. No specific mutation, gene amplification, or previously proposed gene signatures were found to predict sorafenib’s effectiveness. However, several retrospective studies have demonstrated the effectiveness of sorafenib as an adjuvant therapy following surgical resection in preventing recurrence and metastasis in patients with HCC at a high risk of recurrence [11,29].

In a phase II clinical trial that included 31 patients with hepatocellular carcinoma (HCC) and high-risk recurrence factors following radical resection, the administration of sorafenib as an adjuvant therapy showed positive results [30]. The trial compared two groups: one receiving sorafenib adjuvant therapy and the other serving as the control group. The findings indicated that the group receiving sorafenib adjuvant therapy exhibited a longer TTR, with a mean time of 21.45 ± 1.98 months, compared to 13.44 ± 2.66 months in the control group. The difference in TTR between the two groups was statistically significant (*p* = 0.006), indicating that sorafenib treatment delayed recurrence in these patients. Additionally, the sorafenib-treated group had a significantly lower recurrence rate than the control group, with 29.4% and 70.7%, respectively (*p* = 0.032). This suggests that sorafenib adjuvant therapy may reduce the likelihood of recurrence in patients with HCC and high-risk recurrence factors following radical resection. It is worth noting that this was a phase II clinical trial involving a relatively small sample size, and further research with larger cohorts is required to confirm these findings. Additionally, individual patient characteristics, potential side effects, and OS outcomes should also be considered when evaluating the benefits and risks of sorafenib adjuvant therapy in this patient population.

Li et al. [31] investigated the use of sorafenib as an adjuvant therapy within 30 days after surgery in patients with HCC. The findings revealed that patients who received sorafenib treatment experienced an extension of tumor-free survival by approximately 7 months compared to those who underwent surgery alone. This suggests that adding sorafenib to the treatment regimen significantly improved the duration of tumor-free survival. Furthermore, the study reported that the side effects of sorafenib were considered safe and manageable in the patient population. In other words, the use of sorafenib in the adjuvant setting did not result in severe or intolerable adverse effects, supporting its feasibility as a treatment option after surgery for HCC.

In another retrospective study by Wang et al. [32], data from 209 patients with intermediate to advanced HCC at high risk of recurrence after hepatectomy were analyzed. The study compared the outcomes of patients receiving sorafenib with those of the control group. The results demonstrated that the 1-year survival rate was significantly higher in the sorafenib group than in the control group, indicating the potential benefit of sorafenib in improving survival outcomes in these patients. These findings suggest that sorafenib, when administered as an adjuvant therapy after surgery, may play a significant role in prolonging tumor-free survival and improving OS rates in patients with HCC at high risk of recurrence. However, it is important to consider that these studies are retrospective, and further prospective trials are needed to provide more robust evidence and validate these results.

#### 4.1.2. Apatinib

A phase II trial of apatinib [33] involved 30 patients receiving the drug out of 49 screened for postoperative adjuvant treatment of HCC combined with PVTT. This trial was a single-center, open-label, single-arm study with encouraging results, including a 1-year RFS of 36.1% and a 1-year OS of 93.3% following radical hepatectomy. This study suggested that apatinib, in the adjuvant setting, may improve RFS with acceptable tolerability for patients who have undergone resection of HCC with PVTT.

#### 4.1.3. Lenvatinib

The American Society for Clinical Oncology reported interim results from a multicenter, prospective cohort study involving 90 patients with HCC at high risk of recurrence after surgery, demonstrating a significantly longer median DFS in the lenvatinib combined with TACE group compared to the TACE-alone group (12.0 months vs. 8.0 months). The HR was reported as 0.5, indicating a lower risk of recurrence in the group receiving both lenvatinib and TACE [11].

In a study published in *Clinical Medicine Insights: Oncology* in June 2023 by Mu-Gen Dai and colleagues, the impact of adjuvant lenvatinib on survival outcomes in hepatocellular carcinoma (HCC) patients with microvascular invasion (MVI) following curative hepatectomy was assessed [34]. The study included 179 patients, with 43 (24%) receiving adjuvant lenvatinib. Patients who received adjuvant lenvatinib showed significantly improved overall survival (OS) and recurrence-free survival (RFS) compared to those without adjuvant treatment (all *p* < 0.05). This study’s findings suggest the use of oral lenvatinib as an adjuvant therapy in clinical practice for patients with HCC and MVI to reduce tumor recurrence and improve long-term survival.

The studies of the adjuvant target therapy are summarized in Table 2.

### 4.2. Immunotherapy

Atezolizumab and bevacizumab have shown effectiveness as first-line therapies in advanced HCC. Research into tumor immunology has deepened our understanding of the tumor microenvironment (TME) in HCC [38]. The roles and operative mechanisms of immune cells, such as PD-1, CTLA-4, and CD28, in the TME have been elucidated. PD-1 binding to its ligand leads to T cell exhaustion and reduces IFN-γ T cell secretion. CTLA-4 and CD28 mediate immunosuppression by competing for the B7 protein and disrupting the CD28 signal transduction pathway. Various immune checkpoints, including PD-1, PD-L1, and CTLA-4, have been identified in HCC [39]. Clinical trials have been conducted to investigate the biological behavior of these immune checkpoints.

Nivolumab, pembrolizumab, and the combination of nivolumab plus ipilimumab have received FDA approval for the treatment of HCC. Despite the success of immune checkpoint inhibitors (ICIs), there are challenges, including low response rates and side effects. To address the limitations of ICI treatment, researchers are exploring combination therapies.

In a phase II trial [40] assessing the safety and tolerability of perioperative immunotherapy in resectable HCC, 30 patients were enrolled, and 27 of them were randomly assigned to receive either nivolumab alone or nivolumab in combination with ipilimumab. The trial found that the incidence of grade 3–4 adverse events was higher in the group receiving nivolumab combined with ipilimumab than in the group receiving nivolumab alone. Grade 3 and 4 adverse events indicate more severe side effects that may require medical intervention or management. Thus, the combination therapy of nivolumab and ipilimumab led to a higher occurrence of significant adverse events, potentially indicating increased treatment-related toxicity compared to nivolumab monotherapy. However, it is important to consider that adverse events are a known risk with immunotherapy, and these treatments’ overall safety and tolerability profile should be evaluated in the context of potential clinical benefits and further research. Notably, this trial was conducted in a small patient population, necessitating more extensive studies to better understand the safety and efficacy of perioperative immunotherapy in resectable HCC. Additionally, when determining the most appropriate treatment approach, individual patient characteristics and the specific risks and benefits for each patient should be considered. The most common treatment-related adverse events were increased alanine aminotransferase and aspartate aminotransferase levels. No surgery delays occurred due to grade 3 or worse adverse events. Perioperative nivolumab alone and nivolumab plus ipilimumab were deemed safe and feasible for patients with resectable hepatocellular carcinoma. The estimated median progression-free survival (PFS) was longer in the nivolumab plus ipilimumab group compared to nivolumab alone (19.53 months vs. 9.4 months, HR: 0.99, 95% CI: 0.31–2.54).

Patients at a high risk of recurrence, characterized by factors such as increased tumor burden, large or multifocal neoplasms, poor differentiation, and MVI, are likely to benefit the most from adjuvant therapy with ICIs.

The IMbrave050 study [41], is a significant, global, open-label, phase III trial that enrolled adult patients with high-risk surgically resected or ablated HCC from various regions worldwide. The study recruited participants from 134 hospitals and medical centers spanning 26 countries across four WHO regions: the European, American, South-East Asian, and Western Pacific regions.

Patients in the study were randomly assigned in a 1:1 ratio through an interactive voice-web response system, utilizing permuted blocks with a block size of 4. They were allocated to receive either intravenous 1200 mg of atezolizumab in combination with 15 mg/kg of bevacizumab every 3 weeks for 17 cycles (equivalent to 12 months) or to undergo active surveillance without treatment.

The primary endpoint of the Imbrave050 study was RFS, as assessed by an independent review facility in the intention-to-treat population. This primary endpoint aimed to evaluate the efficacy of atezolizumab and bevacizumab combination therapy in preventing disease recurrence in patients with high-risk surgically resected or ablated HCC compared to those on active surveillance. The adjuvant treatment combining atezolizumab with bevacizumab demonstrated a significant improvement in RFS compared to active surveillance in the study. Patients receiving adjuvant atezolizumab plus bevacizumab showed a median RFS that was not evaluable (NE), with a 95% CI ranging from 22.1 to NE, as opposed to the active surveillance group, which also had a median RFS of NE and a 95% CI from 21.4 to NE. The hazard ratio was reported as 0.72, with an adjusted 95% CI of 0.53–0.98, and the statistical significance level was *p* = 0.012.

In terms of safety outcomes, grade 3 and 4 adverse events were more commonly observed in patients who received atezolizumab plus bevacizumab, with 136 out of 332 patients (41%) affected, compared to 44 out of 330 patients (13%) in the active surveillance group. Grade 5 adverse events, which represent severe events leading to death, were reported in six patients (2%, two of whom were related to treatment) in the atezolizumab plus bevacizumab group and one patient (<1%) in the active surveillance group.

Furthermore, the discontinuation of both atezolizumab and bevacizumab due to adverse events was necessary for 29 patients (9%). Managing adverse events and monitoring patient safety remain significant considerations in the utilization of adjuvant atezolizumab plus bevacizumab therapy despite the observed efficacy benefits in improving RFS.

Patients with liver cancer who are candidates for embolization often face high rates of disease progression or recurrence without early access to effective systemic therapies. The EMERALD-1 trial represents a significant advancement in this field. It is a worldwide phase III clinical trial that employed a randomized, double-blind, placebo-controlled approach across multiple centers to evaluate the efficacy of Imfinzi combined with TACE as an initial treatment. This was followed by continued Imfinzi with or without bevacizumab until disease progression, compared to TACE alone, in 616 patients with unresectable HCC suitable for embolization. The study was conducted at 157 medical facilities in 18 countries, including North America, Australia, Europe, South America, and Asia. The primary goal of the trial was to assess PFS in patients receiving Imfinzi and TACE with bevacizumab versus those receiving TACE alone, with secondary endpoints including PFS for patients receiving Imfinzi and TACE, OS, patient-reported outcomes, and objective response rates. Notably, the EMERALD-1 trial stands out as the inaugural global phase III trial, demonstrating enhanced clinical outcomes by combining systemic therapy with TACE to treat liver cancer. It has reported encouraging results, indicating that durvalumab, when combined with TACE and bevacizumab, significantly and meaningfully enhances the primary endpoint of PFS compared to TACE alone in patients diagnosed with HCC suitable for embolization. As the trial progresses, it continues to monitor the secondary endpoint of OS. These findings regarding durvalumab in conjunction with bevacizumab could potentially revolutionize the management of this complex disease, which typically has an unfavorable prognosis, as they mark the first instance where the addition of an immunotherapy combination to TACE has led to a notable improvement in PFS.

Furthermore, an extensive clinical development program is ongoing, which includes investigating the use of Imfinzi in combination with bevacizumab in adjuvant HCC (EMERALD-2) and in combination with tremelimumab, lenvatinib, and TACE for patients eligible for embolization due to HCC (EMERALD-3).

Several ongoing clinical trials are investigating the efficacy of ICIs in the adjuvant setting after loco-regional treatment for HCC, including KEYNOTE-937 (NCT03867084), CHECKMATE 9DX (NCT03383458), and EMERALD-2 (NCT03847428). Table 3 and Table 4 illustrate the ongoing clinical trials of adjuvant ICI therapies for early-stage and intermediate-stage HCC, respectively.

### 4.3. Chemotherapy

Adjuvant chemotherapy for HCC remains a challenging domain, with mixed findings across studies. The unique biological characteristics of HCC, along with the underlying liver disease in patients, contribute to the complexities of achieving consistent and significant therapeutic effects with adjuvant chemotherapy. Further research is needed to refine treatment approaches and identify patient subgroups that may benefit from specific chemotherapy regimens.

#### 4.3.1. Uracil–Tegafur Adjuvant Chemotherapy

The RCT [42] aimed to evaluate the efficacy of postoperative adjuvant therapy with oral uracil–tegafur (UFT) in preventing the recurrence of HCC following curative hepatic resection. The study involved 160 patients randomly assigned to either receive 300 mg/day of UFT for one year after surgery (UFT group) or to undergo surgery alone (control group). Based on the trial results, postoperative adjuvant therapy with oral UFT did not demonstrate significant benefits in preventing HCC recurrence, and there was an indication that it might negatively impact OS.

#### 4.3.2. Oral Capecitabine Adjuvant Therapy

A prospective randomized study [43] conducted between August 2003 and January 2005 in China aimed to evaluate the effectiveness of capecitabine as a postoperative adjuvant regimen in preventing or delaying the recurrence of HCC after surgical resection. The study enrolled 60 patients who underwent hepatectomy for HCC; it found that postoperative adjuvant therapy with oral capecitabine was beneficial in inhibiting the recurrence of HCC, leading to a longer TTR, and potentially improving OS.

#### 4.3.3. Adjuvant Chemotherapy after Liver Transplantation

Studies on the role of systemic chemotherapy following LT for HCC have produced varied results. One study [44] aimed to assess the efficacy of postoperative adjuvant chemotherapy with the FOLFOX regimen on the outcomes of LT for HCC patients who did not meet the Milan criteria. Post-LT adjuvant chemotherapy with FOLFOX did not prevent tumor recurrence post-LT. However, it may improve the survival of HCC patients who do not meet the Milan criteria. The study suggests that adjuvant chemotherapy with the FOLFOX regimen may enhance survival and delay tumor recurrence in HCC patients who do not fulfill the Milan criteria undergoing LT. Another study, which used OXA + 5-Fu + CF adjuvant chemotherapy after transplantation, showed higher 1-, 2-, and 3-year survival rates than those observed in the control group [11].

## 5. Conclusions

This review highlights the challenges in finding globally accepted adjuvant treatment options for postoperative HCC. Various therapies, including TACE, RT, targeted therapy, immunotherapy, and chemotherapy, have been investigated. Although some therapies have shown the potential to improve survival and reduce postoperative recurrence, the lack of strong evidence-based support has hindered the establishment of standard adjuvant treatments for HCC.

One significant aspect of HCC is its association with immune tolerance and escape from immunosurveillance. Consequently, researchers are exploring adjuvant ICIs to enhance the immune response against HCC following curative treatment. These developments in immunotherapy hold promise for improving outcomes in HCC patients.

Combination therapies have also shown encouraging results in advanced HCC. For instance, combining targeted therapy with immunotherapy or TACE has demonstrated positive outcomes. The IMbrave 050 trial has reported favorable results as the first phase III trial of adjuvant treatment for HCC.

An optimal postoperative adjuvant therapy approach should prioritize enhancing the immune system and liver functions while aiming to eliminate residual tumor cells. By targeting the immune system and liver, researchers aim to improve treatment efficacy and reduce the risk of recurrence. However, the specific details and guidelines surrounding these optimized postoperative adjuvant therapies are still evolving, and further research is needed to establish their effectiveness and safety.

In conclusion, no globally accepted standard treatment for HCC is currently available, although adjuvant therapies are being investigated. However, ongoing research into immunotherapy, combination therapies, and optimizing postoperative interventions offer hope for improving outcomes in HCC patients.

## Figures and Tables

**Table 1 cancers-16-01422-t001:** Studies of adjuvant transarterial chemotherapy.

Reference	Study Type	Arms and Intervention	Number of Patients	Main Outcome	Conclusion
**Liu, C., et al. (2016)** [8]	Retrospective study	LR vs. LR + TACE	55 control vs. 62 treatment	Overall: improved 1-year OS with TACE, but no difference in 2- and 3-year DFS rates	For tumor size > 5 cm: improved 1-, 2-, and 3-year DFS. For tumor size ≤ 5 cm: no difference in 1-, 2-, and 3-year DFS
**Ye, J.Z., et al. (2017)** [10]	Retrospective study	LR vs. LR + TACE	260 microvascular invasion (86 in LR +TACE) resection; 259 w/o microvascular invasion (72 in LR + TACE) arm	LR + TACE improved OS and DFS in patients with microvascular invasion but not in patients without microvascular invasion	All patients had BCLC stage A or B
**Liu, Z.H., et al. (2023)** [9]	Retrospective study	LR vs. LR + TACE	421 resected rHCC with MVI-positive patients underwent LR or LR + TACE	Adjuvant TACE provided longer survival for rHCC with MVI when the recurrence time was within 13 months, while not beyond 13 months	For HCC patients with MVI who underwent R0 resection, 13 months may be a reasonable early recurrence time point, and within this interval, postoperative adjuvant TACE may result in longer survival compared with surgery alone
**Chen, W., et al. (2020)** [12]	Meta-analysis	LR vs. LR + TACE	40 studies (10 RCTs and 30 non-RCTs) involving 11,165 patients	PA-TACE was associated with an increased OS and DFS	PA-TACE was beneficial in patients with HCC who were at high risk of postoperative recurrence
**Chen, Z.H., et al. (2019)** [15]	Meta-analysis	LR vs. LR + TACE	12 trials involving 2190 patients	1-, 3-, and 5-year overall survival (OS) rates favored adjuvant TACE over HR alone. Adjuvant TACE showed better 1-, 3-, and 5-year DFS	Adjuvant TACE may improve OS and DFS for HCC patients with MVI

**Table 2 cancers-16-01422-t002:** Studies of adjuvant target therapy.

Reference	Eligible Patients	Arms and Intervention	Number of Patients	Main Outcome	Conclusion
**Xia, F., et al. (2016)** [35]	BCLC-C stage (LR)	Sorafenib vs. no	34 treatment vs. 68 control	The tumor recurrence rate was markedly lower in the sorafenib group (15/34, 44.1%) than in the control group (51/68, 75%, *p* = 0.002). The median disease-free survival was 12 mo in the study group and 10 mo in the control group.	The use of adjuvant sorafenib has been shown to be effective and safe in decreasing hepatocellular carcinoma (HCC) recurrence and extending disease-free and overall survival rates for patients with BCLC-stage C HCC after curative resection.
**Li, J., et al. (2016)** [31]	BCLC-C stage with PVI	Sorafenib vs. no	12 treatment vs. 24 control	The sorafenib group had a significantly longer time to progression (TTP) (29 mo vs. 22 mo, *p* = 0.041) and a significantly longer median OS (37 mo vs. 30 mo, *p* = 0.01).	Patients received sorafenib following surgical resection, was well-tolerated, and demonstrated superior outcomes when compared to those who underwent surgery alone.
**Zhang, X.P., et al. (2019)** [29]	BCLC 0-A or BCLC B with microvascular invasion (MVI)	Sorafenib vs. no	147 treatment vs. 581 control	The overall survival (OS) and recurrence-free survival (RFS) were significantly better for patients in the sorafenib group.	Adjuvant sorafenib was associated with significantly better survival outcomes than LR alone for HCC patients with MVI.
**Bai, S., et al. (2022)** [36]	HBV-related HCC and MVI-positive (LR)	Lenvatinib vs. no	57 treatment vs. 236 control	The 1-year, 2-year recurrence rates, and survival rates of the lenvatinib group were improved compared to the non-lenvatinib group (15.9%, 43.2% vs. 40.1%, 57.2%, *p* = 0.002; 85.8%, 71.2% vs. 69.6%, 53.3%, *p* = 0.009, respectively).	Postoperative adjuvant therapy with lenvatinib was associated with improved long-term prognosis after R0 resection in HBV-related HCC patients with MVI.
**Cai, L., et al. (2020)** [37]	High residual alpha-fetoprotein (LR or ablation)	Lenvatinib vs. TACE vs. no	23 lenvatinib vs. 25 TACE vs. 36 control	61% (14 out of 23) achieved an alpha-fetoprotein (AFP) response in the lenvatinib (LEN) group. The 1-year recurrence-free survival (RFS) rate was notably higher for patients in the LEN group who attained an AFP response at 71.4% (10 out of 14) compared to 36.0% (9 out of 25) in the transcatheter arterial chemoembolization (TACE) group and 50.0% (18 out of 36) in the control group. The median RFS has not been reached in the LEN, TACE, and control groups.	Lenvatinib (LEN) resulted in an AFP response in 61% of HCCs with persistently elevated AFP levels following surgery or ablation. This response was linked to a significantly increased 1-year recurrence-free survival (RFS).

**Table 3 cancers-16-01422-t003:** Ongoing clinical trials: early-stage HCC.

Trial	Test Arm	Comparator	Patient Population	Expected Patients Entry	Primary Endpoint	Trial
**CheckMate 9DX**	Nivolumab	Placebo	High-risk recurrent HCC after radical resection/ablation	530	RFS	NCT03383458
**KEY?NOTE-937**	Pembrolizuamb	Placebo	Imaging CR after surgical resection/local ablation	950	RFS/OS	NCT03867048
**EMERALD-2**	Durvaluamab + bevacitumab	Placebo	High-risk recurrent HCC after radical resection/ablation	888	RFS	NCT03847428

**Table 4 cancers-16-01422-t004:** Ongoing clinical trials: intermediate-stage HCC.

Trial	Test Arm	Comparator	Patient Population	Expected Patients Entry	Primary Endpoint
**LEAP-012**	TACE + lenvatinib + pembrolizuamb	TACE + placebo	Child-Pugh A First treatment (naïve), no extra-hepatic unresectable HCC	950	RFS/OS
**CheckMate-74W**	TACE + ipilimumab + nivolumab	TACE + placebo	Intermediate stage ECOG 0-1 Beyond Milan and up-to-seven criteria	765	Time to TACE progression/OS
**TACE-3**	TACE + nivolumab	TACE alone	Child-Pugh A ECOG 0-1 No extra-hepatic unresectable HCC	522	OS/Time to TACE progression
**TALENT-ACE**	TACE + atezolizumab + bevacizumab	TACE alone	Child-Pugh A ECOG 0-1, untreated TKIs, ICIs	342	TACE PFS/OS
**EMERALD-3**	TACE + durvalumab + tremelimumab +/− lenvatinib	TACE alone	Child-Pugh A ECOG 0-1	525	PFS
**RENO-TACE**	Regorafenib + nivolumab	TACE alone	Beyond up-to-seven criteria	496	PFS
**ABC-HCC**	Atezolizumab + bevacizumab	TACE alone	Child-Pugh A or B7 ECOG 0-1	434	Time to failure of treatment strategy

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
