# Peer review of "The Current and Prospective Adjuvant Therapies for Hepatocellular Carcinoma"

_cancers, 2024, doi:10.3390/cancers16071422_

Round 1

Reviewer 1 Report (Previous Reviewer 2)

Comments and Suggestions for Authors

Much improved manuscript

Author Response

We deeply appreciate the constructive feedback provided by you.

Reviewer 2 Report (Previous Reviewer 3)

Comments and Suggestions for Authors

This is a well written review describing, current therapies in the field of HCC.

Author Response

We deeply appreciate the constructive feedback provided by you.

Reviewer 3 Report (New Reviewer)

Comments and Suggestions for Authors

The review is well written and comprehensive. I think some topics are only superficially covered, for example the authors prepared a long paragraph on immunotherapy and several very short paragraghs on the other drugs. For example the authors should comment more deeply the role of lenvatinib, for example about the comparison over sorafenib (in this regard cite the recent SRMA: PMID: 34017396)

Some figures would improve the quality of the manuscript.

The authors commented a meta-analysis. Maybe they could add a table reporting the main data of the studies included in that meta-analysis or even try to update the results if new studies have been recently published. 

Author Response

We deeply appreciate the constructive feedback provided by you.

  1. The review is well written and comprehensive. I think some topics are only superficially covered, for example the authors prepared a long paragraph on immunotherapy and several very short paragraghs on the other drugs. For example the authors should comment more deeply the role of lenvatinib, for example about the comparison over sorafenib (in this regard cite the recent SRMA: PMID: 34017396)

Response: Thank you for the review’s comment. In accordance with the reviewer's suggestion, we have incorporated reference 27, revised the manuscript from lines 237-248, and added Table 2 to summarize studies on adjuvant target therapy. All revisions are highlighted in red for clarity.

  1. Some figures would improve the quality of the manuscript.(?)

Response: This is a little confusing because there is no Figures in this manuscript.

  1. The authors commented a meta-analysis. Maybe they could add a table reporting the main data of the studies included in that meta-analysis or even try to update the results if new studies have been recently published.

Response: We have included reference 27 as per the reviewer's recommendation. The manuscript has been revised accordingly, and Table 2 has been added to summarize studies on adjuvant target therapy.

We confirm that the contents of this manuscript have not been copyrighted or published previously, nor are they under consideration for publication elsewhere. All authors have actively contributed to the planning, execution, or data analysis of the study and have approved the final version submitted. Additionally, the manuscript has been reviewed by an experienced editor fluent in English, specializing in editing papers authored by non-native English speakers. There are no conflicts of interest with any financial organization regarding the material discussed in the manuscript.

We thank you once again for your invaluable feedback and guidance throughout this process.

Round 2

Reviewer 3 Report (New Reviewer)

Comments and Suggestions for Authors

The revised version of the manuscript is OK. Thank you!

This manuscript is a resubmission of an earlier submission. The following is a list of the peer review reports and author responses from that submission.

Round 1

Reviewer 1 Report

Comments and Suggestions for Authors

This review article provides an update on current and prospective adjuvant therapy for HCC. It is due to be published by the end of 2024.

The authors list the various ongoing trials, as well as the first results published or communicated.

With regard to intra-arterial approaches, we currently have concordant and interesting data in the adjuvant situation. However, these studies are mainly based in Asia, and probably involve non-cirrhotic patients. The transposability of these results is therefore difficult.

As regards systemic treatments, few data are currently available (STORM, IMBRAVE 050 preliminary results, not mature for OS).

The bulk of the results are still to come, and in particular the results of EMERALD 1 and 2, the mature data from IMBRAVE 050 and possibly the results of the 9DX study and Keynote 937 are due to be published in 2024. It therefore seems inappropriate not to address these major results for changing our practices in a review article on current and prospective adjuvant therapy for HCC. For example, the results of the EMERALD 1 trial (positive press release) will be released by the end of January...

I have two other problems with the publication of this article.

Regarding the issues addressed, we absolutely must have a discussion on :

- The inclusion criteria for the various studies (size, number, surgery, radiofrequency, etc.).

- The perspectives between pure adjuvant situations (post-surgery, radiofrequency), combined treatments (with TACE or SIRT) and neoadjuvant approaches, which seem very promising (with the need to list ongoing neoadjuvant trials).

Reviewer 2 Report

Comments and Suggestions for Authors

This is a fine and comprehensive review of current and future therapies for HCC

My comments:

Overall, the English should be improved.

For each of the 3.1, 3.2 paragraphs you should consider adding a table. That will ensure overall better understanding of the text

Abbrevations: the same abbreviations are explained again and again multiple times (DFS, OS, HCC, PVC et cetera). Explain the abbreviation once only. And maybe include a footnote with all abbreviations

Comments on the Quality of English Language

The quality is overall poor. I suggest collaborating with an English-speaking person

Reviewer 3 Report

Comments and Suggestions for Authors

In this manuscript the authors present evidence about the current treatment regimes of  HCC patients (pre operative and post operative) in relation to recurrence. 

The present manuscript is a decent review of the literature, but it needs some English edditing especialy in the "Simple summary" and the "Abstract" sections.  

Comments on the Quality of English Language

As previously mentioned.